# Possessing potential weapons (still) heightens anger perception: Replicating and extending a test of error management theory

Cody Moser[1]*, Richard Ellk[2], Ayonna Jones[2], Colin Holbrook[2]

1 School of Collective Intelligence, Université Mohammed VI Polytechnique, Rabat, Morocco,
2 Department of Cognitive and Information Sciences, University of California Merced, Merced, California, United States of America

* cmoserj@gmail.com

**Data availability statement:** All data files including the pre-registration, study materials, analysis script in R, and results from the survey

## Abstract

Error Management Theory (EMT) hypothesizes that humans are functionally biased to err on the side of the least costly mistake when making judgments under uncertainty. Applying EMT to emotion perception, previous studies found that people perceive individuals holding everyday objects with potentially lethal affordances to be both higher in state anger and more anger-prone, relative to individuals holding control objects. Here, we conduct a direct replication of one such study examining these effects [13] and exploratorily test whether friends of an individual possessing a potentially lethal tool are also perceived as angrier—and hence more dangerous—in a "reverse-halo" effect. Participants (N = 476) were presented an image of an individual depicted as an avid cook and either holding or near a conventional kitchen knife, then asked to rate their emotional states and trait dispositions. Participants were next presented with an image of a second individual framed as a close friend of the cooking enthusiast and asked to assess their emotional states and dispositions. The results replicated prior findings that individuals depicted holding everyday objects with incidentally lethal affordances are perceived as angrier, more prone to anger, and more socially unpleasant. However, we did not find evidence for a "reverse-halo" effect extending to friends. Instead, all of the state and trait emotion ratings of the two individuals were significantly correlated, consistent with an inference (orthogonal to error management motivations) of homophilic similarity in the affective profiles of friends. These results are discussed as they inform prior EMT research and motivate further study of the determinants of emotion perception.

## Introduction

When faced with incomplete or uncertain information, decision-makers risk two kinds of errors: over-estimations of the information available (leading to false positives, or Type I errors) and under-estimations of the information available (leading to false negatives, or

are all archived on OSF at
https://osf.io/4c2ue/?view_only=
84954626f5954b43bb7af78a2c690ac9.

**Funding:** The author(s) received no specific funding for this work.

**Competing interests:** The authors have declared that no competing interests exist.

Type II errors). When the costs associated with Type I versus Type II errors are asymmetrical, Error Management Theory (EMT) holds that, under uncertainty, individuals should be biased towards the least costly error [1,2]. Within human cognition, EMT encompasses the perception of emotions, which can provide highly fitness-relevant cues of the future behavior of others. Emotion perception is inherently imprecise, in part because individuals are often motivated to mask their own emotional states. When assessing the emotional state of an individual, EMT predicts that their capacity to inflict harm or confer benefits may be expected to bias emotion-perception toward the least costly error [3]. For example, all else being equal, the greater the individual's capacity to inflict harm, the more costly underestimation of their anger becomes [4]. Consistent with this model, prior research found that target individuals presented as holding objects with incidentally lethal affordances (i.e., garden shears or a kitchen knife), were estimated as more angry, and more anger-prone, relative to when presented holding contextually equivalent yet non-lethal control objects (i.e., a gardening can or spatula). This effect has replicated both directly [4] and conceptually in meta-analytic examinations [5].

In addition to perceptions of the emotions of individuals depicted in isolation, a growing literature is beginning to highlight the role that EMT plays in assessing the emotions of individuals depicted in the context of groups [6]. For example, individuals tend to misclassify crowds as more angry than individuals, particularly in more ambiguous contexts, such as when faces are presented with lower intensities of expression [7,8]. Insofar as access to allies enhances the perceived formidability of potential antagonists [9], these effects complement findings that individuals with access to objects which could be leveraged as weapons are perceived to be angrier. Relatedly, previous work on "halo effects" and "horn effects" indicates that positive or negative assessments of individuals embedded within social groups are influenced by assessments of other group members, although these dynamics have yet to be explored with regard to emotion perception [10,11].

Here, we sought to integrate the above lines of inquiry by i) replicating prior research into the impact of possession of potentially dangerous tools on perceived anger, while ii) simultaneously exploring whether the hypothesized error management effects on appraisals of target individuals extend to appraisals of their allies. Do participants deem a person as being more angry when they are framed as a friend of someone equipped with a potentially dangerous object? To explore the potential for such a "reverse-halo" effect, we posed parallel questions regarding attributions of state and trait emotions to a target individual and to an individual framed as their friend. In addition, in another exploratory extension of prior work, we collected assessments of the envisioned bodily size and strength of the friend character, as such ratings can index estimates of the danger posed by potential antagonists [12]. To the extent that a target individual's friend is appraised as potentially dangerous, they should be envisioned as both relatively angry and physically formidable.

## Materials and methods

Adult participants were recruited from throughout the United States via Prolific.co in exchange for $0.75. Because we did not know the effect size which may obtain with regard to the anticipated effect of the weapon manipulation on appraisals of the friend character, and because we conducted a direct replication of elements of Holbrook et al. ([13], Study 1), we simply doubled the sample size used in that study. To evaluate the adequacy of our sample size, a post-hoc power analysis was conducted for the nine dependent variables representing ratings made for the friend across two conditions. The analysis was based on a small effect size ($f^2$ = .02). The results of the post-hoc power analysis indicate that a total sample size of 784

observations is needed to achieve adequate statistical power for detecting a small effect. Per our pre-registration criteria, duplicated participants, smartphone users, individuals with -2.5 SD of duration time in the experiment, and those who failed one of two catch questions were excluded. Following exclusions, the prior study included 264 participants; we therefore sought 600 participants in the current study during the dates of May 16–24, 2022, obtaining 626 at the end of data collection, yielding a final sample of 476 (median age = 34; 64% women, 32% men, 4% other; 78% White).

Replicating the between-subjects design used by Holbrook et al. ([13], Study 1), following provision of written informed consent, participants initially viewed a single photo of a man described as enjoying cooking in his everyday life. In the Armed condition, the model was digitally manipulated to be posed holding a kitchen knife. The model is depicted holding the knife out to his side, which might be interpreted as having been perceived as implicitly threatening, and thereby confounding the lethal affordance of this every day item with an aggressive motive in the model. We do not believe that this interpretation plausibly explains our findings, for two reasons. First, participants were informed that the models had been asked to pose for a photo while holding an object intended to illustrate their hobby, hence the model's stance should not have conveyed threatening intent, but rather cooperatively presenting their hobby-relevant object to the camera, as asked. Second, in the original study, the same model was compared while holding different objects (a spatula, garden shears, and a flowercan) in the exact same stance, and the objects lacking lethal affordances elicited significantly distinct effects on state or trait emotion attribution relative to the knife or shears. In the Unarmed condition, the same model was posed empty-handed, with the kitchen knife depicted in a separate window displayed adjacent to the photograph of the model. This presentation held constant the visual features of the stimuli, including potential semantic associations with violence which might be evoked by the image of a knife, while making it clear that the target individual in the unarmed control condition was not presently holding the knife. The unarmed control model's arm was placed at his side to avoid the impression that their hand was balled in a raised fist when depicted without the grasped object, potentially connoting anger or violent intent (see Fig 1). Participants were asked to rate the model's degree of state anger, fear, and disgust on a scale rating (0 = *Not at all*; 9 = *Extremely*), as well as trait anger, fear, disgust, dishonesty, and unpleasantness (0 = *much less than average*; 9 = *much more than average*). The state and trait questions were presented in separate blocks (randomized), with the order of question items randomized within blocks. The photograph and questions about the target were visible simultaneously.

Next, immediately following ratings of the target, participants viewed a photo of a neutral male face (taken from the Radboud Faces Database, Langner et al. [14]) described as a close friend of the previous individual and were asked to estimate the same state and trait emotions (see Fig 1). We selected three white male faces to portray the ostensible friend character to hold the apparent race/ethnicity of the target and the friend constant, and thereby avoid potential confounding inferences related to group stereotypes. As before, the state and trait questions were presented in separate blocks (randomized), with the order of items randomized within blocks. The photograph and questions about the target were visible simultaneously. Participants were then asked to estimate the friend's height (in feet and inches) and to rate their size and muscularity using arrays depicting, respectively, silhouettes differing only in size, and bodies differing only in muscularity. Following prior research (e.g., Fessler et al. [4]), these three measures of envisioned bodily formidability (stature, size, and muscularity) were standardized and averaged to create a composite (a = 0.72).

Participants were then asked two exploratory questions about their overall degree of confidence in their assessments of both individuals: "Overall, how confident did you feel

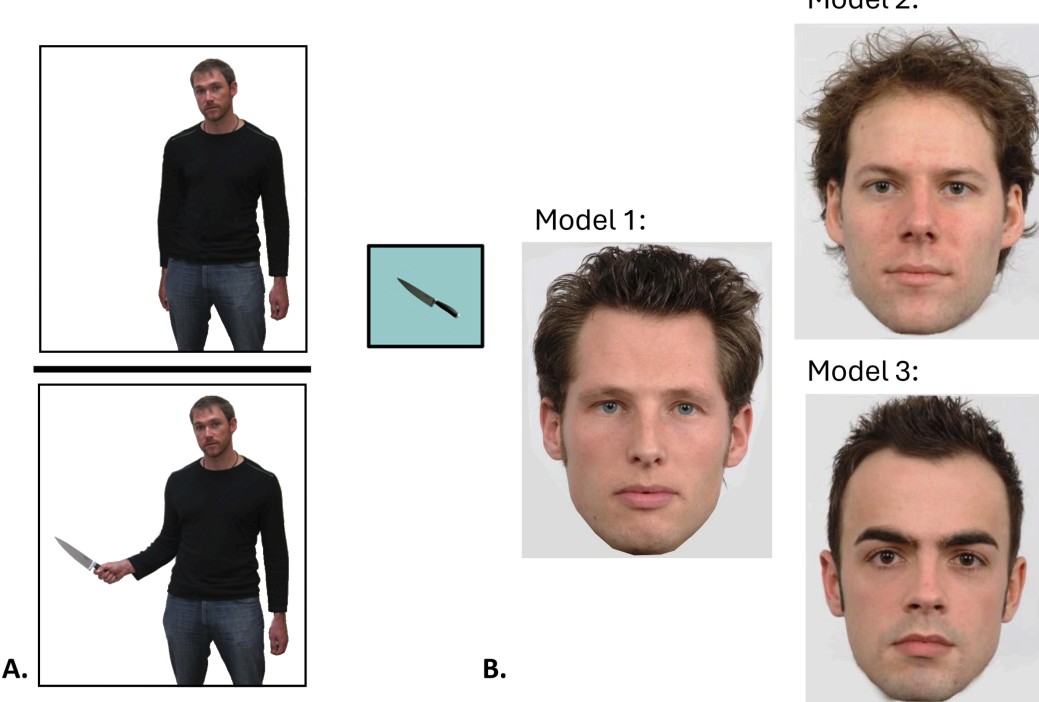

**Fig 1. Stimuli representing Target and Friend.** Each participant first viewed a man described as enjoying cooking. In the Unarmed condition (Panel A, top) he was depicted adjacent to a kitchen knife; in the Armed condition (Panel A, bottom) he was depicted holding the kitchen knife. Next, participants viewed one of three neutral male faces described as a close friend of the first man (Panel B).

when rating the feelings and personality traits of the first man you saw [the close friend (the second man you saw)]?" (1 = *Not confident at all*; 9 = *Completely confident*). Finally, participants answered demographic questions and were debriefed.

## Pre-registered hypotheses

We pre-registered several predictions regarding replication of Holbrook et al.'s [13] findings as well as novel predictions concerning the potential "reverse-halo" or "horn" effect:

As in Holbrook et al. [13], relative to the target character in the Unarmed condition depicted not holding a knife, we predicted that the target holding a knife would be estimated to be:

1. Higher in state anger
2. Higher in trait anger
3. Lower in trait fear
4. Lower in trait disgust
5. Higher in trait dishonesty
6. Higher in trait unpleasantness

Relative to the friend of the target character in the Unarmed condition depicted not holding a knife, we predicted that the friend of the target holding a knife would be estimated to be:

1. Higher in state anger
2. Higher in trait anger
3. Lower in trait fear
4. Lower in trait disgust
5. Higher in trait dishonesty
6. Higher in trait unpleasantness
7. Higher in envisioned physical formidability

Note that we pre-registered the latter prediction with respect to envisioned physical formidability as contingent on the extent to which the friend character was perceived to be relatively dangerous (operationalized here according to how angry they were appraised to be), anticipating that potential effects on bodily estimates may be quite modest given the relatively weak and indirect connection to danger (i.e., alliance with a friend who happens to have access to a conventional kitchen knife). Indeed, the relatively weak connection to danger cues inherent to this manipulation (e.g., framing the character as a close friend of a cook rather than a violent criminal) renders the predictions with regard to state and trait emotion as well as envisioned physical formidability highly conservative tests of the impact of EMT considerations to social appraisals.

All statistical analyses were conducted in R Team et al. [15]. To assess the effects of weapon manipulation on state and trait ratings of the target and the friend, we employed two separate general linear models on the target and friend measures in the form of MANOVAs to account for simultaneous multiple dependent variables. Separate from the target model, in the friend model we control for which friend's face was displayed. To assess the possibility that, for reasons orthogonal to EMT, participants would infer homophily between the target and his friend, we similarly ran correlations of the state and trait ratings between the two individuals [16,17]. All preregistrations, data, scripts, and materials are available at https://osf.io/4c2ue/.

## Results

### Replication of Holbrook et al. [13]

As an initial step, we examined descriptive statistics for all emotional and trait ratings by condition (see Table 1), followed by independent samples $t$-tests to assess differences between the armed and unarmed conditions (see S1 Table for full statistical results, including Cohen's $d$). We identified significant differences in state ($M_{armed}$ = 4.36, SD = 2.31; $M_{unarmed}$ = 3.64, SD = 2.00) and trait anger ($M_{armed}$ = 5.36, SD = 1.79; $M_{unarmed}$ = 5.02, SD = 1.61) of the target, as well as in trait unpleasantness ($M_{armed}$ = 4.98, SD = 1.76; $M_{unarmed}$ = 4.31, SD = 1.49). These effects were statistically significant: state anger, $t(457)$ = 3.64, $p < .001$, $d = 0.34$; trait anger, $t(460)$ = 2.19, $p = .029$, $d = 0.20$; and trait unpleasantness, $t(453)$ = 4.44, $p < .001$, $d = 0.41$. All effect sizes fell within the small to approaching moderate range.

Prior to conducting the MANOVA analyses, we evaluated whether the assumptions of multivariate normality and homogeneity of covariance matrices were met. Mardia's test indicated significant violations of multivariate normality for both the target and friend rating variables (Target: skewness $\chi^2$ = 380.50, $p < .001$; kurtosis $z$ = 9.02, $p < .001$; Friend: skewness $\chi^2$ = 725.75, $p < .001$; kurtosis $z$ = 13.56, $p < .001$). However, Box's M test for equality of covariance matrices revealed no significant difference in covariance structures across experimental groups for the target ratings ($p = .141$), suggesting that the assumption of homogeneity was met. In contrast, Box's M test for the friend ratings was significant ($p = .0003$), indicating that the homogeneity of covariance matrices assumption may be violated in that set. While MANOVA is generally robust to moderate departures from multivariate normality and

**Table 1. Descriptive statistics for emotion and trait ratings by armed condition.**

| State/Trait | Emotion | Model | Mean (Armed) | SD (Armed) | Mean (Unarmed) | SD (Unarmed) |
|---|---|---|---|---|---|---|
| State | Anger | Target | 4.36 | 2.31 | 3.64 | 2.00 |
| State | Anger | Friend | 3.60 | 2.00 | 3.43 | 2.05 |
| State | Fear | Target | 3.72 | 2.09 | 3.60 | 1.99 |
| State | Fear | Friend | 3.25 | 1.94 | 3.15 | 1.83 |
| State | Disgust | Target | 3.05 | 1.92 | 3.14 | 1.88 |
| State | Disgust | Friend | 2.84 | 1.73 | 2.80 | 1.80 |
| Trait | Anger | Target | 5.36 | 1.79 | 5.02 | 1.61 |
| Trait | Anger | Friend | 4.62 | 1.81 | 4.71 | 1.58 |
| Trait | Fear | Target | 3.73 | 1.66 | 3.61 | 1.49 |
| Trait | Fear | Friend | 3.76 | 1.67 | 3.88 | 1.58 |
| Trait | Disgust | Target | 3.49 | 1.68 | 3.46 | 1.48 |
| Trait | Disgust | Friend | 3.63 | 1.56 | 3.80 | 1.58 |
| Trait | Dishonesty | Target | 4.54 | 1.60 | 4.32 | 1.39 |
| Trait | Dishonesty | Friend | 4.54 | 1.59 | 4.69 | 1.51 |
| Trait | Unpleasantness | Target | 4.98 | 1.76 | 4.31 | 1.49 |
| Trait | Unpleasantness | Friend | 4.63 | 1.75 | 4.59 | 1.62 |
| Trait | Formidability | Target | -0.02 | 0.73 | 0.02 | 0.72 |

Means and standard deviations of emotion and trait ratings by condition, model, and emotion type. Ratings were on a 7-point scale (except Formidability, which was a z-scored composite measure).

unequal covariances, particularly with balanced group sizes and sufficient sample size, these assumption checks should be considered when interpreting the results.

With respect to the MANOVA, we observed significant differences between the armed and unarmed conditions in state anger ($F(1, 461) = 13.29$, $p < .001$, $\eta^2 = 0.03$) and trait anger ($F(1, 461) = 5.62$, $p = .018$, $\eta^2 = 0.01$), indicating support for these predictions (see Table 2). Against our other predictions, we found no significant differences for trait fear, disgust, or dishonesty and these exhibited very small or negligible effect sizes (see Table 2 and Fig 2 to visualize rating distributions). In support of our final prediction, the Armed target was rated more unpleasant than the Unarmed target ($F(1, 461) = 19.80$, $p < .001$, $\eta^2 = 0.04$). These effects correspond to small effect sizes, suggesting reliable but modest differences in perceived anger based on the presence of a weapon. Importantly, despite assumption violations, the MANOVA results mirrored the pattern observed in the t-tests, with the same three outcome variables—state anger, trait anger, and trait unpleasantness—emerging as statistically

**Table 2. Effects of armed condition on perceived state and trait emotions.**

| State/Trait | Emotion | $F$ | $\eta_p^2$ | $p$ |
|---|---|---|---|---|
| State | Anger | 13.29 | 0.03 | <0.001* |
| State | Fear | 0.48 | 0.00 | 0.491 |
| State | Disgust | 0.26 | 0.00 | 0.612 |
| Trait | Anger | 5.62 | 0.01 | 0.018* |
| Trait | Fear | 0.71 | 0.00 | 0.400 |
| Trait | Disgust | 0.04 | 0.00 | 0.842 |
| Trait | Dishonesty | 2.83 | 0.01 | 0.093 |
| Trait | Unpleasantness | 19.80 | 0.04 | <0.001* |
| Trait | Confidence | 0.36 | 0.00 | 0.550 |

Results from linear regressions predicting state and trait emotional ratings of the target based on possession versus non-possession of a knife. Partial eta squared ($\eta_p^2$) reflects standardized effect size. * Indicates p < .05

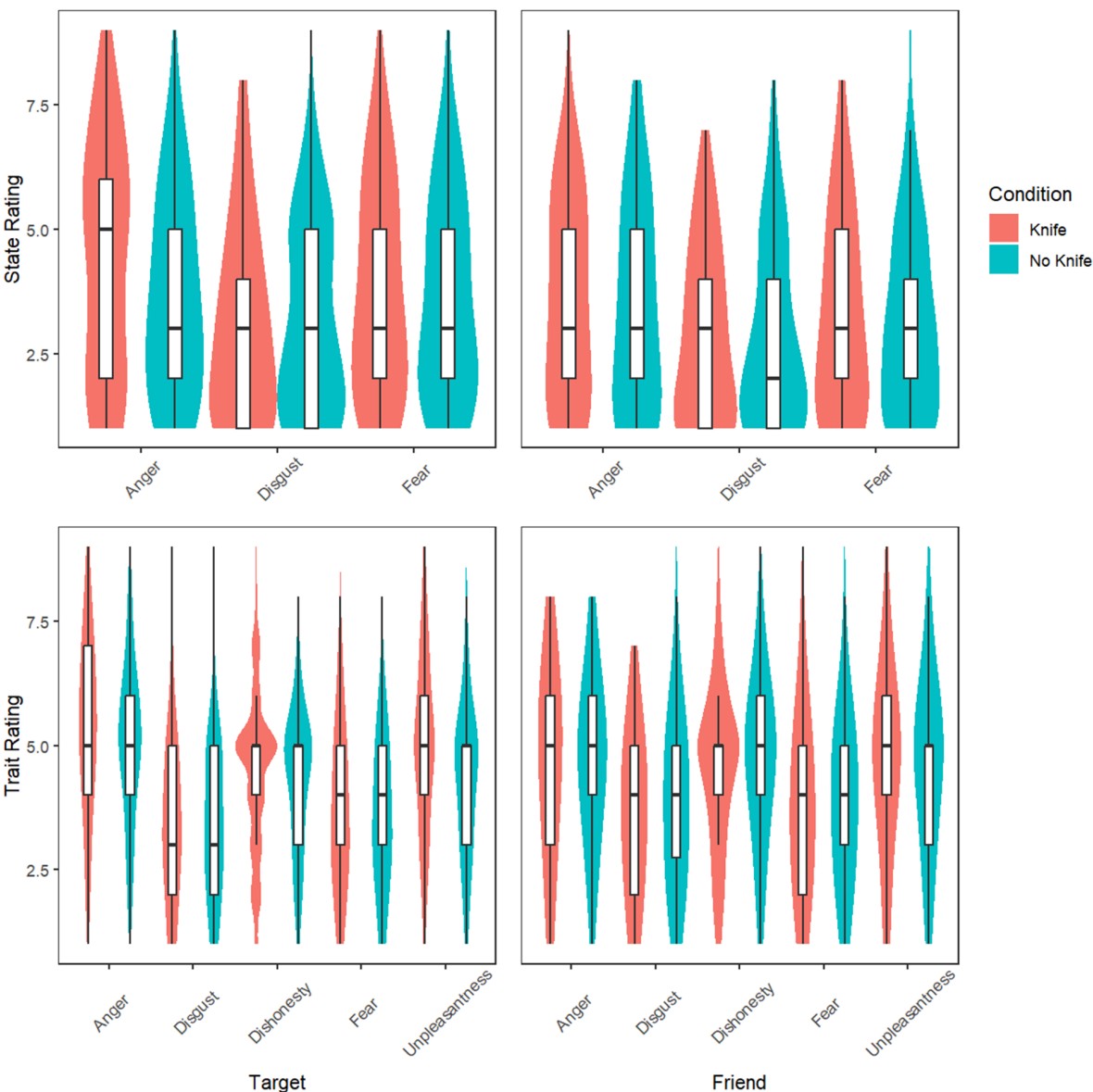

**Fig 2. State and trait ratings for target and friend.** Top Row: Ratings by participants of emotional states of both the target individual (left) and his friend (right) dependent on whether the friend was presented holding a knife or without a knife. Bottom Row: Ratings by participants of emotional traits of target individual (left) and his friend (right) dependent on the same condition as above.

significant. This convergence of results across analytical approaches increases confidence in the robustness of the observed effects.

### Tests of a potential "reverse halo" effect

We ran multiple linear regressions on friend state and trait anger with both target state and trait anger and condition as our predictor variables. Against our predictions for the friend, we found no effects of condition on ratings of the friend character (see Table 3). With regards to the friend state and trait anger and trait unpleasantness, there were strong effects for the

**Table 3. Null effects of armed condition on perceived state and trait emotions of friend.**

| State/Trait | Emotion | F | $\eta_P^2$ | p |
|---|---|---|---|---|
| State | Anger | 0.58 | 0.00 | 0.447 |
| State | Fear | 0.46 | 0.00 | 0.499 |
| State | Disgust | 0.00 | 0.00 | 0.954 |
| Trait | Anger | 0.21 | 0.00 | 0.645 |
| Trait | Fear | 0.24 | 0.00 | 0.627 |
| Trait | Disgust | 0.84 | 0.00 | 0.360 |
| Trait | Dishonesty | 0.87 | 0.00 | 0.352 |
| Trait | Unpleasantness | 0.16 | 0.00 | 0.689 |
| Trait | Formidability | 0.05 | 0.00 | 0.825 |
| Trait | Confidence | 0.07 | 0.00 | 0.788 |

Results from linear regressions predicting state and trait emotional ratings, formidability, and confidence of the friend based on presence or absence of the target wielding a knife. Partial eta squared ($\eta_P^2$) reflects standardized effect size.

model. We conducted post-hoc Tukey comparisons for the three friend models across both conditions, finding that Model 3 was rated to be significantly higher in state and trait anger than both Models 2 and 3, while the two themselves did not differ (S2 Table). After analyzing participants which only received Models 1 or 2, our results for ratings of the friend remained robust (S3 Table).

Fig 2 shows the results from both the state and trait ratings of both the target and friend, by condition. As can be seen most, means are similar for each variable, with the exception of state anger and traits anger and unpleasantness for the target.

## Tests of inferred homophily between target and friend

In line with our exploratory test of the possibility that participants would infer affective homophily between friends, we found highly significant linear positive correlations (*r*s 0.29–0.48, *p*s <.001) between the two characters for all state and trait measures (Table 4).

## Discussion

We found that possession of an everyday object with incidentally lethal affordances heightened perception of state and trait anger, as well as trait unpleasantness, consistent with EMT

**Table 4. Correlations between target and friend state and trait ratings.**

| State/Trait | Emotion | r | p |
|---|---|---|---|
| State | Anger | 0.33 | <0.001 |
| State | Disgust | 0.36 | <0.001 |
| State | Fear | 0.29 | <0.001 |
| Trait | Anger | 0.34 | <0.001 |
| Trait | Disgust | 0.39 | <0.001 |
| Trait | Fear | 0.32 | <0.001 |
| Trait | Dishonesty | 0.48 | <0.001 |
| Trait | Unpleasantness | 0.40 | <0.001 |

Values reflect Pearson correlations between friend and target state and trait emotional ratings, as a test of homophily; e.g. in the case of State Anger, the correlation reflects the association between the target's and the friend's state anger ratings.

and replicating previously observed effects (Holbrook et al. [13]; also see Fessler et al. [4]). However, departing from prior findings, we observed no effects of object manipulation on trait fear, disgust, or dishonesty in our study. This discrepancy may reflect a more accurate estimate of effect sizes provided by the larger sample (N = 476) compared to the initial study (N = 264). The absence of effects in the larger sample may therefore indicate a more precise estimate of these effects and that the original findings, based on a smaller sample, could have been overestimated. While similar hypotheses—such as semantic priming and the "general aggression model" [5]—suggest that associations with weapons should heighten general aggression tendencies, these effects were ruled out by controls in the original Holbrook et al. [13] study. Fessler et al. [4] replicated similar findings using both a knife and garden shears, further examining the relationship between lethal and non-lethal affordances in assessments of a target's formidability. The overall pattern of results supports the prediction that dangerous object possession up-regulates perception of anger and related traits in a manner consistent with an adaptive capacity to minimize risk of potentially costly aggression.

Extending prior research, we tested whether heightening perceived anger due to framing an individual as possessing a potentially lethal object would similarly heighten perceptions of the anger or physical formidability of a friend. We found, contrary to predictions, that the object manipulation had no effect on any of the appraisals of the friend. While this null finding may be interpreted as evidence against the proposed "reverse-halo" effect with respect to appraisals of allies, three methodological limitations should be noted. First, while we doubled the size of the sample from the original Holbrook et al. [13] study, a post-hoc power analysis indicated that a sample size of 784 observations would be necessary for detecting a small effect size, compared to the 476 obtained in this study. Second, whereas prior research examining the emotional assessment of members of groups have presented individuals together [6–8], our experiment presented the two individuals sequentially, potentially minimizing conceptual associations between the target person and their friend, including the specific impression that should the armed individual attack, their friend would be local enough to be able to join them. Third, the online approach employed here inherently lacks ecological validity. Indeed, given that the individual holding a knife is depicted using a mere photograph on a computer screen–and hence not likely to pose any real-world danger to the participant–it is remarkable that observers reliably perceive this person as relatively angry and unpleasant in character. Were a person possessing a potentially lethal object and accompanied by an ostensible friend encountered in a lab space (e.g., using research confederates), or encountered as characters in a realistic simulation (e.g., using augmented or virtual reality), friends might indeed be perceived as relatively angry, unpleasant, and formidable along with the person holding the object. Relatedly, the friend stimuli, presented in the current study as one of three cropped neutral faces (done so as to assess intuitions regarding their bodily formidability), were inherently less ecologically valid than the target stimuli, and may not have evoked emotion appraisals comparably to the presentation of the target's full body. Thus, although the present study does not provide support for a "reverse-halo" effect on anger perception, future work employing more valid methods is necessary.

Finally, in an exploratory test of the extent to which friends would be inferred to possess similar emotional profiles, we examined whether the target's trait and state emotions correlated with those of the friend. Indeed, consistent with inferences of affective homophily and prior research [16,17], and despite the lack of an effect of manipulation on ratings of the friend, we observed strong correlations between all of the state and trait ratings. These positive results motivate future work on the role of inferred homophily in emotion perception and detection.

In closing, we successfully replicated previously documented shifts in attributions of state anger and related antisocial traits (i.e., anger-proneness, social unpleasantness) caused by manipulation of a highly subtle contextual determinant: the target individual's possession of an ordinary kitchen knife in their hand versus ownership of the same knife in an unspecified location. Although we have focused here on effects of dangerous object possession on perception of threat-relevant emotions and traits, EMT predicts analogous effects of subtle context cues on functional biases in emotions and traits relevant to the potential conferral of benefits (e.g., romantic interest, career opportunities). We encourage affective scientists to draw upon EMT when generating hypotheses regarding contextual determinants of emotion categorization.

## Supporting information

**S1 Table. Independent samples T-tests and Cohen's _d_ for Armed vs. Unarmed conditions.** Independent samples t-tests comparing emotional and trait ratings between armed and unarmed conditions, disaggregated by rater. Cohen's _d_ is included as a standardized measure of effect size. * Indicates p < .05.
(PDF)

**S2 Table. Post-hoc Tukey T-tests for state and trait anger between friends.** Results from post-hoc Tukey post-hoc HSD test examining the effect of model on state and trait anger. Note significant differences between Model 3 and Models 1 and 2, which do not differ from one another. * Indicates p < .05.
(PDF)

**S3 Table. Null effects of armed condition on perceived state and trait emotions of friend after the removal of model 3.** Results from linear regressions predicting state and trait emotional ratings and formidability of the friend based on presence/absence of the target wielding a knife, excluding Model 3.
(PDF)

## Acknowledgments

We thank Alejandro Pérez Velilla for feedback on the study during its initial design phase.

## Author contributions

**Conceptualization:** Cody Moser, Colin Holbrook, Richard Ellks, Ayonna Jones.

**Formal analysis:** Cody Moser.

**Funding acquisition:** Colin Holbrook.

**Investigation:** Cody Moser.

**Methodology:** Cody Moser, Colin Holbrook, Richard Ellks, Ayonna Jones.

**Project administration:** Colin Holbrook.

**Supervision:** Colin Holbrook.

**Visualization:** Cody Moser.

**Writing – original draft:** Cody Moser.

**Writing – review & editing:** Cody Moser, Colin Holbrook, Richard Ellks.

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
