## [Decision Letter · Decision Letter 0]

15 Aug 2024

PONE-D-24-25340Possessing Potential Weapons (Still) Heightens Anger Perception: Replicating and Extending a Test of Error Management TheoryPLOS ONE

Dear Dr. Moser,

Thank you for submitting your manuscript to PLOS ONE. After careful consideration, we feel that it has merit but does not fully meet PLOS ONE’s publication criteria as it currently stands. Therefore, we invite you to submit a revised version of the manuscript that addresses the points raised during the review process.

Title: "Possessing Potential Weapons (Still) Heightens Anger Perception: Replicating and Extending a Test of Error Management Theory":

**Clarity Issues**: The title is somewhat convoluted and might be difficult for readers to immediately grasp. The phrase "Possessing Potential Weapons (Still)" is particularly vague and could be clearer. It's not immediately obvious what "Still" refers to—whether it's the persistence of a previous finding or a temporal aspect.**Overly Complex**: The title tries to convey too much information in a single line. It combines replication, extension, and the application of Error Management Theory, which could be confusing. Simplifying the title to focus on the primary aspect of the research might make it more accessible.**Lack of Specificity**: The title doesn’t specify what kind of "potential weapons" are being discussed. This could lead to ambiguity about the scope and context of the study. Specifying whether the research focuses on physical weapons, symbolic threats, or another form of potential aggression would improve clarity.**Redundancy**: The phrase "Heightens Anger Perception" could be seen as redundant in the context of "Error Management Theory," which may already imply a focus on perception and response. This might suggest that the title could be more concise.**Jargon Heavy**: The use of terms like "Error Management Theory" might be too technical for a broader audience. While this may be appropriate for a specialized academic journal, it could limit the title's appeal and accessibility to a wider readership.**Potential for Misinterpretation**: The title might be interpreted as suggesting a direct causative relationship between possessing potential weapons and heightened anger perception without adequately explaining the mechanisms or variables involved. This could lead to misunderstandings about the study’s findings.**Lack of Novelty Indication**: The title mentions "Replicating and Extending," but doesn’t clearly indicate what new insights or contributions the research offers beyond previous studies. It might benefit from highlighting what novel aspect or new understanding the study brings to the field.

****

Here are some alternative titles that might address the issues with the original title while maintaining the core focus of the research:

**"The Impact of Potential Weapon Possession on Anger Perception: New Insights from Error Management Theory"****"How Potential Weapons Influence Anger Perception: A Replication and Expansion of Error Management Theory"****"Reevaluating Anger Perception in the Presence of Potential Weapons: Insights from Error Management Theory"****"Potential Weapons and Anger Perception: A Study Extending Error Management Theory"****"The Role of Potential Weapons in Heightening Anger Perception: A Test of Error Management Theory"****"Exploring the Link Between Potential Weapons and Anger Perception: A Replication of Error Management Theory"****"Does Possessing Potential Weapons Affect Anger Perception? A New Test of Error Management Theory"****"Potential Threats and Anger Perception: An Updated Analysis of Error Management Theory"**

**Abstract:**

**Lack of Novelty**: The abstract highlights that the study replicates previous findings without introducing significant new insights or contributions. The mention of a "reverse-halo" effect and its failure to be observed might not be sufficiently compelling if the main contribution is merely confirming past results.**Complex Language**: The abstract uses technical jargon and complex phrasing that might not be accessible to a broader audience. Terms like "affordances," "orthogonal to error management motivations," and "homophilic similarity" could be confusing for readers unfamiliar with the specific theoretical framework.**Unclear Motivation for Study**: The abstract doesn’t clearly justify why exploring the “reverse-halo” effect is important. The motivation for testing this effect and its implications for Error Management Theory (EMT) could be better articulated to underscore its relevance.**Methodological Details**: While the abstract describes the general methodology, it lacks details on how the images were presented or the specific nature of the control objects used. More information on these aspects could help in assessing the validity and reliability of the findings.**Inadequate Discussion of Results**: The results are summarized but not deeply analyzed. The abstract notes the replication of previous findings and the absence of a “reverse-halo” effect, but it does not delve into possible reasons for these outcomes or their implications for EMT.**Overemphasis on Negative Findings**: The failure to observe the “reverse-halo” effect might overshadow the importance of the replicated findings. The abstract could benefit from a more balanced discussion that highlights the significance of the replicated results alongside the null findings.**Lack of Specific Implications**: The abstract mentions that the results “inform prior EMT research” and “motivate further study” but does not specify how. Providing concrete examples of how these findings could influence future research or practical applications would strengthen the abstract.**Sample Details**: The abstract mentions a sample size of 476 participants but does not provide information on participant demographics or how they might affect the study's generalizability. Including such details could enhance the understanding of the study's context and limitations.

**Introduction section:**

**Overuse of Jargon**: The introduction uses specialized terms and concepts, such as "Error Management Theory," "Type I and Type II errors," and "affordances," which may be unclear or confusing to readers not familiar with these concepts. Simplifying the language or providing brief definitions could improve readability.**Lack of Clear Focus**: The introduction attempts to cover a wide range of topics, from basic Error Management Theory to the specifics of emotion perception in groups and halo effects. This breadth might make the introduction seem unfocused and could dilute the primary research question or objective.**Insufficient Background Context**: While the introduction mentions previous research and related concepts, it does not provide enough background on why the specific study is important or how it fills a gap in the literature. More emphasis on the significance of the research question would strengthen the context.**Unclear Motivation for Study**: The rationale for exploring the “reverse-halo” effect in the context of individuals with potentially dangerous objects could be better articulated. The introduction should make it clearer why this specific extension of the research is necessary and how it advances the field.**Complexity of Sentence Structure**: Some sentences are complex and lengthy, which can hinder understanding. For example, sentences that introduce multiple concepts or studies simultaneously could be broken down into simpler, more digestible parts.**Redundancy**: There is some repetition of ideas, particularly concerning the impact of possessing potentially dangerous tools and the effects of context on emotion perception. Streamlining these sections to avoid redundancy could make the introduction more concise and focused.**Insufficient Definition of Key Terms**: Key terms like "reverse-halo effect" and "error management effects" are mentioned but not clearly defined or explained. Providing definitions or a brief explanation of these concepts would help readers grasp the research's context better.**Limited Discussion of Previous Research**: Although the introduction references prior studies, it does not critically assess them or discuss their limitations. A more detailed review of previous findings and their shortcomings could highlight the need for the current study more effectively.**Inadequate Transition to Research Questions**: The transition from discussing background information to stating the research questions is abrupt. A smoother transition that clearly connects the background information to the specific aims of the study would improve coherence.**Lack of Specificity in Hypotheses**: The introduction outlines broad research aims but does not specify detailed hypotheses or predictions. Providing clear, specific hypotheses would give readers a better understanding of what the study intends to test.

**Materials and Methods section:**

**Inadequate Justification for Sample Size**: The decision to double the sample size from the prior study is mentioned, but the rationale behind this choice is not fully explained. More detail on how the sample size was determined, including power analysis or estimates of effect size, would strengthen the justification.**Recruitment Compensation**: Offering $0.75 for participation may be perceived as insufficient compensation, which could affect the quality of responses and participant motivation. This low payment might lead to concerns about the adequacy of participant engagement and data quality.**Lack of Diversity in Sample**: The sample is predominantly White (78%), which may limit the generalizability of the findings. There is no mention of how this demographic composition might impact the results or the steps taken to address potential biases related to ethnicity.**Insufficient Detail on Exclusion Criteria**: While the introduction of exclusion criteria is noted (e.g., duplicated participants, smartphone users), there is limited information on the specific criteria used to identify these issues. More detail on how these criteria were applied and their impact on the final sample would be useful.**Ambiguity in Experimental Conditions**: The description of the Armed and Unarmed conditions could benefit from additional detail. For example, it’s unclear how the digital manipulation was performed or whether there were any potential confounds associated with the images.**Potential for Demand Characteristics**: The simultaneous presentation of the photograph and rating questions might lead to demand characteristics, where participants’ responses are influenced by the immediate visual context. This could affect the validity of the reported emotions.**Limited Description of Control Conditions**: The control condition involves showing a knife in a separate window, but it is not clear if the visual presentation of the knife might still influence participants’ perceptions or if the knife image was entirely neutralized.**Exploratory Measures**: The exploratory questions about confidence in assessments might not be sufficiently justified or linked to the primary research aims. These questions could be seen as extraneous if they do not clearly contribute to understanding the main hypotheses.**Potential Bias in Face Selection**: The use of three white male faces to represent the friend character aims to avoid racial or ethnic confounding. However, this choice could still introduce biases related to race or ethnicity that are not addressed in the analysis or discussion.**Unclear Statistical Methods**: The section mentions the standardization and averaging of measures but does not provide details on the statistical methods used for analysis. More information on how the data were analyzed, including any statistical tests or models, would be beneficial.**Methodological Limitations Not Addressed**: There is no discussion of potential limitations related to the methodology, such as the potential impact of digital manipulation on the realism of the stimuli or the generalizability of the findings.**Confusion About Ratings**: The introduction of rating scales for both state and trait emotions is clear, but the specific rationale for including both types of ratings and how they contribute to the research objectives could be better explained.

**Pre-registered Hypotheses section:**

**Redundancy and Lack of Clarity**: The hypotheses related to the friend of the target character are very similar to those for the target character, with only minor variations. This redundancy could make the section seem repetitive and may not clearly justify why these specific predictions about the friend character are necessary. The explanation could be more concise and focused on the novelty of the hypotheses.**Insufficient Justification for Predictions**: While the hypotheses are listed, there is limited justification or theoretical grounding provided for why these specific predictions are made. The introduction to the predictions could benefit from a clearer explanation of the underlying rationale or theoretical framework that supports these expectations.**Ambiguity in Contingent Predictions**: The mention that predictions regarding envisioned physical formidability are contingent on the friend character being perceived as dangerous could be confusing. The explanation for this contingency is vague, and it is unclear how this might affect the interpretation of the results.**Potential Overemphasis on Weak Effects**: The section notes that the predictions are "highly conservative" due to the "relatively weak connection to danger cues." This might be seen as an attempt to preemptively justify the lack of strong effects, which could be perceived as a way to downplay potential negative results or lack of significant findings.**Lack of Specificity in Statistical Analysis**: The description of the statistical methods is minimal and lacks detail. For instance, the choice of a general linear model is mentioned, but there is no explanation of how variables were controlled for, or why this particular model was chosen over other possible analyses. More detail on the statistical approach and its appropriateness for the hypotheses would be beneficial.**Insufficient Discussion of Potential Limitations**: The pre-registered hypotheses section does not address any potential limitations of the study design or hypotheses. A discussion of possible limitations or challenges in testing these hypotheses would provide a more balanced perspective.**No Mention of Effect Size Considerations**: The section does not discuss expected effect sizes or how they were estimated. Including this information would help in understanding the anticipated magnitude of the effects and the adequacy of the sample size.**Lack of Integration with Prior Research**: The section does not clearly integrate the pre-registered hypotheses with existing research beyond the immediate context. Providing more context on how these hypotheses build upon or diverge from prior work would strengthen the rationale.**Unclear Operationalization of Constructs**: The operationalization of constructs like "envisioned physical formidability" is mentioned but not thoroughly explained. Clear definitions and how these constructs are measured would improve the clarity and replicability of the study.**Insufficient Detail on the Friend Face Model Control**: While it is noted that the analyses control for which friend face model was displayed, there is no explanation of how this control was implemented or how it might impact the results. More detail on this aspect of the analysis would be useful.

**Results section:**

**Insufficient Detail on Non-Significant Results**: The section reports a lack of significant differences in several traits (e.g., trait fear, disgust, dishonesty) without providing much detail on these non-significant findings. It would be useful to include more information about effect sizes and power analyses to better understand the implications of these null results. Without this, it is difficult to gauge whether the non-significant results are due to true absence of effect or insufficient power.**Inconsistencies in Reporting**: The results mention "strong effects" for the friend model but do not clearly define what constitutes a "strong effect" in this context. The use of subjective terms like "strong effects" could be replaced with more precise statistical measures or effect sizes to ensure clarity and objectivity.**Lack of Explanation for Post-Hoc Analyses**: The post-hoc Tukey comparisons and subsequent results are mentioned briefly without a clear explanation of why these analyses were necessary. Including a rationale for performing post-hoc tests and how they contribute to the understanding of the data would enhance the clarity and robustness of the results section.**Ambiguous Interpretation of “Reverse-Halo” Effect**: The results section states that no effects of condition were found on ratings of the friend character but does not delve into potential reasons for this null finding beyond the methodological limitations. More detailed discussion of why the hypothesized "reverse-halo" effect might not have been observed could provide deeper insights.**Insufficient Context for Correlations**: The significant correlations found between state and trait ratings of the target and friend are reported, but the implications of these correlations are not fully explored. More discussion is needed on how these correlations contribute to the understanding of affective homophily and whether they support or contradict the hypotheses.**Methodological Limitations Briefly Noted**: The discussion of methodological limitations, such as the sequential presentation of the target and friend and the online nature of the study, is somewhat superficial. A more thorough exploration of how these limitations specifically impact the findings and suggestions for addressing these issues in future research would strengthen the section.**Overemphasis on Significant Results**: There is a noticeable focus on the significant findings related to state and trait anger and unpleasantness, while the non-significant results receive less attention. Balancing the discussion to equally address and interpret non-significant results would provide a more comprehensive view of the data.**Lack of Integration with Pre-Registered Hypotheses**: The results section could better integrate with the pre-registered hypotheses by clearly mapping out which predictions were supported or refuted. A more direct comparison between the hypothesized and observed results would help in evaluating the study's contribution to the existing literature.**Presentation of Statistical Data**: The tables and figures are mentioned but not thoroughly discussed. Providing more detailed explanations of the statistical data presented in Tables 1 and 2, as well as Figures 2, would enhance the reader's understanding of the results.**Exploratory Findings**: The exploratory findings on inferred homophily are mentioned but not thoroughly analyzed. A more detailed discussion on how these findings contribute to the overall research question and their implications for future studies would be beneficial.

**Discussion section:**

**Lack of Critical Analysis of Null Results**: The discussion briefly mentions the null results for traits such as fear, disgust, and dishonesty but does not deeply explore the potential reasons for these findings. The lack of significant results is mentioned as potentially due to increased statistical power, but more detailed analysis and potential explanations for the absence of expected effects are needed.**Methodological Limitations**: Although methodological limitations are acknowledged, such as the sequential presentation of individuals and the use of online platforms, the discussion does not sufficiently address how these limitations might have specifically influenced the findings. A more detailed critique of how these factors could have impacted the results and suggestions for overcoming these limitations in future research would provide a clearer understanding of the study's limitations.**Overemphasis on Replication Success**: The discussion focuses heavily on the successful replication of certain effects, particularly related to state and trait anger and unpleasantness. While replication is important, the discussion could benefit from a more balanced view that also critically examines the aspects where the study did not replicate prior findings.**Insufficient Exploration of “Reverse-Halo” Effect**: The absence of support for the "reverse-halo" effect is mentioned but not explored in depth. The discussion should more thoroughly investigate why the hypothesized reverse-halo effect was not observed and consider alternative explanations or factors that may have influenced this outcome.**Exploratory Findings and Their Implications**: The exploratory findings on affective homophily are briefly noted but not discussed in depth. A more detailed interpretation of how these findings contribute to the understanding of emotion perception and the implications for future research would strengthen the discussion.**Generalizability Issues**: The discussion touches on the lack of ecological validity of the online approach but does not elaborate on how this might limit the generalizability of the findings. More emphasis on how the study's results might differ in real-world settings or using more immersive research methods would be useful.**Future Research Directions**: While future research is mentioned, the discussion could provide more specific recommendations for subsequent studies. For instance, what exact aspects of the "reverse-halo" effect need further exploration, and how can future research address the limitations identified in this study?**Implications of Findings**: The discussion mentions the implications of the findings for Error Management Theory (EMT) and affective science but does so in a somewhat cursory manner. A deeper analysis of how the findings specifically advance or challenge existing theories and what this means for practical applications or theoretical advancements would be beneficial.**Inconsistent Emphasis**: The emphasis on certain results, like the significant findings of state and trait anger, might overshadow other important aspects of the study. A more balanced discussion that equally addresses both significant and non-significant results, and integrates these into the broader context of the study's aims, would provide a more comprehensive view.**Clarity and Structure**: The discussion could benefit from clearer structuring. It jumps between various topics, such as replication success, methodological limitations, and exploratory findings, without a clear narrative thread. A more organized discussion that systematically addresses each aspect of the study and ties them back to the research questions would improve readability and coherence.

**References:**

**Reference Duplication:****Issues:**References 4 and 5 are identical, creating unnecessary duplication in the reference list. This redundancy can confuse readers and detract from the professionalism of the manuscript.
**Recommendation:** Remove the duplicated reference and ensure each reference entry is unique.
**Incomplete or Incorrect Details:****Issues:**Reference 2 is incomplete, lacking details such as the full journal name or volume and page numbers.Reference 16 is cited as “RTeam. Core. R,” which is not a standard citation format and does not provide sufficient details for readers to locate the source.
**Recommendation:** Ensure all references are complete and correctly formatted according to the journal’s citation style. For example, provide full details for software references, including the version number and accurate publication information.
**Outdated or Irrelevant Sources:****Issues:**Some references, like Remmers and Martin (1944), may be outdated given the field's advancements in understanding the halo effect.Several references are from niche or less widely-read journals, which might limit the broader applicability of the cited works.
**Recommendation:** Consider including more recent and widely-recognized sources to ensure the literature review reflects current research trends and standards.

**Comments on Supplementary Tables:****Lack of Contextual Information:****Issues:**Tables 4, 5, and 6 lack adequate context or explanation within the supplementary materials. Readers may struggle to understand the relevance of these tables without additional narrative or background information.
**Recommendation:** Provide a brief description or explanation for each supplementary table to clarify their significance and how they relate to the main findings of the study.
**Data Interpretation and Clarity:****Issues:**Table 4’s post-hoc results are presented with limited interpretation, making it difficult to understand the implications of the findings without further explanation.Table 5 and Table 6 focus on the removal of Model 3, but the rationale for this removal and its impact on the results are not clearly explained.
**Recommendation:** Include a discussion or summary that interprets the results shown in these tables, particularly how the removal of Model 3 affects the overall conclusions.
**Inconsistency in Reporting:****Issues:**The p-values and F-values in Tables 5 and 6 are reported without accompanying effect sizes or confidence intervals, which are crucial for interpreting the practical significance of the results.
**Recommendation:** Report effect sizes and confidence intervals in addition to p-values and F-values to provide a more comprehensive view of the data’s implications.
**Potential Errors or Omissions:****Issues:**The p-values for some tests in Tables 5 and 6 are very close to conventional significance thresholds (e.g., p = 0.054), which may suggest potential issues with the data or analysis that require further scrutiny.
**Recommendation:** Re-evaluate the data and ensure that all statistical analyses are accurately reported and interpreted. Address any borderline results with a detailed explanation of their implications.

**Statistical Concerns and Results Interpretation:****Table 4: Post-Hoc Tukey T-Tests****Significant Differences:** The Tukey HSD test results reveal significant differences between Model 3 and Models 1 and 2 for state anger and trait anger. The p-values for contrasts between Model 1 and Model 3 and Model 2 and Model 3 are all significant, indicating that Model 3 differs significantly in its impact on state anger and trait anger compared to the other models. However, the non-significant results for other comparisons (e.g., State M1-M2, Trait M1-M2) might suggest that these models do not differ meaningfully in certain aspects.**Implications:** These results suggest that Model 3 has a different effect compared to Models 1 and 2. The study needs to better address why Model 3 shows such a pronounced difference and what this means for the overall findings. The variation across models needs clearer interpretation and discussion.
**Table 5: Effects After Removal of Model 3****Effect Sizes and p-Values:** The F-values and p-values for the effects of the armed condition on various emotions show that while state anger and trait unpleasantness are significantly affected, trait anger's p-value is on the borderline of significance (p = 0.054). This indicates potential issues with the robustness of some findings.**Implications:** The removal of Model 3 changes the significance of some effects. The study must clarify how the exclusion of a model impacts the results and ensure that the findings are consistently reliable.
**Table 6: Null Effects on Friend’s Emotions****Null Findings:** The results indicate null effects for most emotions (e.g., state anger, trait anger, trait fear) when the armed condition is present. This raises questions about the lack of impact on the friend’s emotional ratings and the generalizability of the findings.**Implications:** The null results for the friend’s emotional perceptions suggest that the study’s hypothesis about the “reverse-halo” effect may not be supported. This needs a thorough discussion of why these null results occurred and what it means for the study’s conclusions.

2. **Discussion and Theoretical Implications:****Results and Hypotheses:** The tables reveal discrepancies between predicted and observed effects. For instance, the "reverse-halo" effect hypothesis is not supported by the data. The discussion needs to address why the expected effects did not materialize and how this impacts the theoretical framework.**Ecological Validity:** The study’s findings might be influenced by the experimental design, including the use of photographs and online methods. The discussion should provide a more robust explanation of how these factors might affect the results and their practical implications.

3. **Reference and Supplementary Materials:****Accuracy and Completeness:** The references and supplementary materials must be reviewed for accuracy and completeness. Any inconsistencies or errors in data presentation can affect the study's credibility and require correction.**Clarity in Reporting:** The presentation of supplementary tables should be clear and directly related to the main results. Any discrepancies or unexpected results should be addressed and explained in the context of the main findings.

This revision should involve:**Re-evaluating Statistical Analyses:** Ensure all models are appropriately assessed and clarify the impact of removing Model 3.**Clarifying Theoretical Implications:** Address how the findings align or deviate from theoretical predictions and discuss the implications for the proposed “reverse-halo” effect.**Improving Results Presentation:** Ensure accuracy and clarity in reporting results and references.

Thank you for your attention to these matters.

In addition to addressing these editor’s comments, please ensure that you also incorporate the suggestions from the reviewers. Please submit your revised manuscript by Sep 29 2024 11:59PM. If you will need more time than this to complete your revisions, please reply to this message or contact the journal office at plosone@plos.org. Please include the following items when submitting your revised manuscript:

We look forward to receiving your revised manuscript.

Kind regards,

Pradeep Paraman

Academic Editor

PLOS ONE

Reviewers' comments:

Reviewer's Responses to Questions

**Comments to the Author**

1. Is the manuscript technically sound, and do the data support the conclusions?

Reviewer #1: Yes

Reviewer #2: Yes

Reviewer #3: Yes

2. Has the statistical analysis been performed appropriately and rigorously? 

Reviewer #1: Yes

Reviewer #2: Yes

Reviewer #3: Yes

3. Have the authors made all data underlying the findings in their manuscript fully available?

Reviewer #1: Yes

Reviewer #2: Yes

Reviewer #3: Yes

4. Is the manuscript presented in an intelligible fashion and written in standard English?

Reviewer #1: Yes

Reviewer #2: Yes

Reviewer #3: Yes

5. Review Comments to the Author

Reviewer #1: Where in USA was this research carried out?This is not stated. A brief description of the study site (population and ethnic composition included) will assist us apppreciate the study conditions and also agree/or not on the appropriateness of the study sample

Reviewer #2: Review of: Possessing Potential Weapons (Still) Heightens Anger Perception: Replicating and

Extending a Test of Error Management Theory

MSID: PONE-D-24-25340

This manuscript professes to replicate previous findings about Error Management Theory (EMT) and to extend the findings to examine potential “halo-effects” to neutral associated individuals. In sum, I find the research design and empirical evaluations to be acceptable (with a possible exception), and the results sound given the experiment.

First and foremost, this is not my area of expertise. While I am competent to evaluate the research design and implementation, this literature is foreign to me, and I do not mean to suggest otherwise in my review. With this caveat noted, here are my impressions. First, I am a bit surprised to see the results of this experiment being expressed in terms of linear regression models. Given the authors’ use of Latex and R, I would have expected a more sophisticated modeling schema, especially given the truncated data and non-linear comparisons. Conversely, dealing with experimental design in this manner, I would have expected to see the discussion of and the results presented of a Latin Square design. Not that this matters much, as the OLS models will replicate the more conventional ANOVA presentation – it's just interesting.

I see no deficits in the empirical modeling nor in the interpretation of that modeling. The authors do an excellent job of qualifying the findings. Similarly, I have no problems with the sampling strategy or sample size. My concerns with this manuscript are twofold. First, the treatment illustration portrays an individual with a chef’s knife in a partially extended arm. My concern here is that attitudinal data of perceived threats extends beyond inanimate objects to body position and posturing. Given that this is a somewhat atypical posture for a “cooking enthusiast” to hold a knife, I have concerns about the validity of the neutral armed image. Secondly, and far less important, I am skeptical that this replication presents much in the way of substantive contribution to the literature – but that is a question far better suited for someone with better knowledge of the literature.

Reviewer #3: The target manuscript reports the results of one study. Its primary goal is to replicate the results of Halbrook et al (2014, study 1), with a secondary goal of exploring the so called "reverse-halo effect". The manuscript is very well written (I found only a typo in page 3, line 82: the sentence ends with quotation marks), clear and rigorous while also managing to be engaging – it was truly a pleasure for me to read this work. I also congratulate the authors for their effort to replicate an important result – despite its widely acknowledged relevance, it is unfortunate that most authors seldomly devote their efforts to replications.

In regards to the secondary goal – exploring the "reverse-halo effect" – there is room for improvement and I could easily suggest some ways to strengthen the methodological choices. However, I will abstain from doing so in this review. For one, it is unlikely that any of my suggestions would supplant the authors’ own perspectives for future studies. On the other hand, within the logic of this manuscript, where that part of the study is manifestly exploratory in nature, any such possible modifications of the methodology might compromise the main goal. All things considered, I found the work reported in this manuscript to be well thought and well balanced, given its two aims (the thoughtful discussion on limitations greatly contributes to this assessment).

Maybe the only section that I found could be slightly improved is the “Materials and methods”. On a first reading, it was not immediately clear to me what parts of the task were similar when assessing the Target and the Friend – only when reading the results did I understand that most questions regarding the Target were also posed for the Friend. If the authors decide to revise this section, it would only improve an already great paper.

In sum, I found this manuscript to be as close to be published in its current from as possible. For that, I congratulate the authors.

6. PLOS authors have the option to publish the peer review history of their article (what does this mean?). If published, this will include your full peer review and any attached files.

Reviewer #1: **Yes:** Willice O. Abuya

Reviewer #2: **Yes:** Warren S Eller

Reviewer #3: **Yes:** Nuno Alexandre De Sá Teixeira

---

## [Author Response · Author response to Decision Letter 1]

10 Oct 2024

Replies to Editor

As the recommendations appear to have been generated with the use of a Large Language Model, many of the suggestions lacked contextually relevant understanding of the hypotheses, literature or scholarly norms. In our revision, accordingly, we have primarily focused on addressing the feedback generated by the reviewers. However, we also carefully considered the AI-generated feedback, and did find some substantive suggestions which we address below.

- Suggestion of an alternative title

One suggestion was that the current title may be too jargon-laden. The alternative titles suggested, were either so broad as to not convey the key findings of the study (e.g. “"The Role of Potential Weapons in Heightening Anger Perception: A Test of Error Management Theory"), or were actually lengthier and more jargony than the current title (e.g. “"How Potential Weapons Influence Anger Perception: A Replication and Expansion of Error Management Theory"). We concluded that the current title, which straightforwardly states the findings of the current study and straightforwardly identifies it as a replication, is more appropriate.

- Suggestion of a simpler abstract

Another line of feedback was that the abstract, as it is written, does not emphasize the study’s novelty and that we overemphasize the null findings. PLOS-ONE’s mission statement as a journal prioritizes replicable, open science over novelty or asymmetrically highlighting significant effects over nulls. Indeed, we identified PLOS as a suitable venue for this project in part because the key significant findings replicate previously published work. While not novel, this sort of work is indispensable for cumulative science to proceed. Therefore, we do not believe that it would be appropriate or necessary to minimize our null results or reframe the results in a way that would attempt to exaggerate novelty. It was also suggested that details of our participant sample be provided directly in the abstract, whereas scholarly conventions dictate that such details are left to readers in the text.

- Introduction of terminology in Introduction Section

In the introduction section, the AI argues that we have overused jargon and not clearly defined our key terms, such as Error Management Theory and Type I and Type II errors. To the contrary, these terms are defined in the first two sentences (lls. 2-6) before they are employed in the text.

“When faced with incomplete or uncertain information, decision-makers risk two kinds of errors: over-estimations of the information available (leading to false positives, or Type I errors) and under-estimations of the information available (leading to false negatives, or Type II errors). When the costs associated with Type I versus Type II errors are asymmetrical, Error Management Theory (EMT) holds that, under uncertainty, individuals should be biased towards the least costly error (Haselton & Nettle, 2006; Haselton & Galperin, 2012).”

- Methods

The AI stated that we should clarify our sample size and exclusion criteria, which were clarified in lls. 50-55 of the text.

“Because we did not know the effect size which may obtain with regard to the anticipated effect of the weapon manipulation on appraisals of the friend character, and because we were replicating elements of Holbrook et al. (2014, Study 1), we simply doubled the sample size used in that study. Per our pre-registration criteria, duplicated participants, smartphone users, individuals with -2.5 SD of duration time in the experiment, and those who failed one of two catch questions were excluded.”

Additionally, bias in face selection (that we selected three white males faces) was brought up as a concern, which we addressed in lls. 91-93 of the text.

“We selected three white male faces to portray the ostensible friend character to hold the apparent race/ethnicity of the target and the friend constant, and thereby avoid potential confounding inferences related to group stereotypes.”

- Results

The primary concern regarding our results was the effect of Model 3. In the manuscript (lls. 153-158), this is discussed at length, and we analyze our results with and without these Models in SI Tables 1, 2, and 3. Similarly, while the feedback states we did not describe what is controlled for in our linear model, this is explicitly stated in lls. 141-143 of the text.

- Discussion

Finally, the feedback notes 1) cognitive demands and 2) a lack of ecological validity in our approach. These issues are acknowledged and explored in a lengthy paragraph (lls. 183-205).

- Reference Issues

As reference 2 was incomplete, we have updated our reference list. Additionally, as references 4 and 5 were duplicated, we have removed one.

Haselton M. G., Galperin A. Error management and the evolution of cognitive

bias. In: Social Thinking and Interpersonal Behavior. Psychology Press; 2013. p.

45-63

Replies to Reviewer 1

R1.1: “Where in USA was this research carried out? This is not stated. A brief description of the study site (population and ethnic composition included) will assist us appreciate the study conditions and also agree/or not on the appropriateness of the study sample.”

We thank Reviewer 1 for their comment on our paper. In the original version of the manuscript, we state on line 49 that participants were recruited online via Prolific. As we did not want to violate privacy concerns in our dataset and believed our sample size was large enough to capture a wide enough distribution of participants across the United States, we did not collect location data from participants in this study. Nevertheless, we believe our online sample size likely captures a wide variety of Americans who participate in studies online.

To clarify our recruitment for the reasons given by the Reviewer, we have added additional text (ll. 49) to the Materials and methods stating that adult participants were recruited from throughout the United States.

“Adult participants were recruited from throughout the United States via Prolific.co in exchange for $0.75.”

Replies to Reviewer 2

R2.1: “This manuscript professes to replicate previous findings about Error Management Theory (EMT) and to extend the findings to examine potential “halo-effects” to neutral associated individuals. In sum, I find the research design and empirical evaluations to be acceptable (with a possible exception), and the results sound given the experiment.”

We thank Reviewer 2 for their work in reviewing our paper and generous appraisal of our approach.

R2.2: “First and foremost, this is not my area of expertise. While I am competent to evaluate the research design and implementation, this literature is foreign to me, and I do not mean to suggest otherwise in my review. With this caveat noted, here are my impressions. First, I am a bit surprised to see the results of this experiment being expressed in terms of linear regression models. Given the authors’ use of Latex and R, I would have expected a more sophisticated modeling schema, especially given the truncated data and non-linear comparisons. Conversely, dealing with experimental design in this manner, I would have expected to see the discussion of and the results presented of a Latin Square design. Not that this matters much, as the OLS models will replicate the more conventional ANOVA presentation – it's just interesting.”

We believe that the Reviewer’s interpretation of the analysis we conducted in our study may be due to a difference between disciplines. We agree with the Reviewer that this question could have been approached from a range of different statistical models - in particular, using a signal detection approach with binary logistic regression. However, we chose to utilize conventional ANOVA and linear regression models to facilitate comparison to the results of the original Holbrook et al. study. Nevertheless, we agree with the author that other models might have been used and would have conceptually replicated our results.

R2.3: “I see no deficits in the empirical modeling nor in the interpretation of that modeling. The authors do an excellent job of qualifying the findings. Similarly, I have no problems with the sampling strategy or sample size. My concerns with this manuscript are twofold. First, the treatment illustration portrays an individual with a chef’s knife in a partially extended arm. My concern here is that attitudinal data of perceived threats extends beyond inanimate objects to body position and posturing. Given that this is a somewhat atypical posture for a “cooking enthusiast” to hold a knife, I have concerns about the validity of the neutral armed image.”

The Reviewer raises the ecological concern that our treatment condition of the target holding a knife may be more threatening in appearance based on the target’s stance rather than on the object being held. While we agree this could pose a confound for the experiment, in the sense that the target may be primed, we note that in the original study, the model was presented with several additional objects in the same stance (as these objects were photoshopped in): a spatula, garden shears, and a flowercan. In this case, only the image with the knife or the garden shears (both objects with lethal affordances) were seen as threatening by participants. In addition, the framing of the photograph to participants explained that the models had been asked to pose for a photo while holding an object illustrating their hobby, rather than while performing their hobby (i.e., cooking). As such, it would make sense for the model to hold and display the object in this way for the photo. We thus believe that the condition with the knife in our study likely replicates the original findings while serving as an ideal priming mechanism for our extension examining the influence (or, in our case, lack thereof) that the target has on ratings of a friend. In the revision, we have added an acknowledgment of this issue as a potential limitation, reminded readers of the framing explaining why a cooking enthusiast would hold the knife in this way for the photograph, and described how this potential confound was handled in the original paper (i.e., by utilizing alternate objects held with arm extended) (lls. 62-73).

“The model is depicted holding the knife out to his side, which might be interpreted as having been perceived as implicitly threatening, and thereby confounding the lethal affordance of this every day item with an aggressive motive in the model. We do not believe that this interpretation plausibly explains our findings, for two reasons. First, participants were informed that the models had been asked to pose for a photo while holding an object intended to illustrate their hobby, hence the model’s stance should not have conveyed threatening intent, but rather cooperatively presenting their hobby-relevant object to the camera, as asked. Second, in the original study, the same model was compared while holding different objects (a spatula, garden shears, and a flowercan) in the exact same stance, and the objects lacking lethal affordances elicited significantly distinct effects on state or trait emotion attribution relative to the knife or shears.”

R2.4: “Secondly, and far less important, I am skeptical that this replication presents much in the way of substantive contribution to the literature – but that is a question far better suited for someone with better knowledge of the literature.”

We disagree that our study does not make a contribution to the literature for two reasons. First, in terms of the replication, the original Holbrook study, at 10 years old, has been due for a replication given recent failures in the priming literature, a method which is suggested by the methodology of this study. Perhaps more to the point, novelty should not be considered a more valuable aspect of research published in PLOS than replicability, as one of the reasons the journal was created was to encourage replication projects in service of cumulative, reliable science.

Replies to Reviewer 3

R3.1: The target manuscript reports the results of one study. Its primary goal is to replicate the results of Halbrook et al (2014, study 1), with a secondary goal of exploring the so called "reverse-halo effect". The manuscript is very well written (I found only a typo in page 3, line 82: the sentence ends with quotation marks), clear and rigorous while also managing to be engaging – it was truly a pleasure for me to read this work. I also congratulate the authors for their effort to replicate an important result – despite its widely acknowledged relevance, it is unfortunate that most authors seldomly devote their efforts to replications.

We thank the reviewer for their generous sentiments regarding our study and current version of the manuscript, as well as for their close reading of the text. We have updated the error the author found on page 3 and removed the quotation marks. We agree that replication is invaluable!

R3.2: “In regards to the secondary goal – exploring the "reverse-halo effect" – there is room for improvement and I could easily suggest some ways to strengthen the methodological choices. However, I will abstain from doing so in this review. For one, it is unlikely that any of my suggestions would supplant the authors’ own perspectives for future studies. On the other hand, within the logic of this manuscript, where that part of the study is manifestly exploratory in nature, any such possible modifications of the methodology might compromise the main goal. All things considered, I found the work reported in this manuscript to be well thought and well balanced, given its two aims (the thoughtful discussion on limitations greatly contributes to this assessment).

Maybe the only section that I found could be slightly improved is the “Materials and methods”. On a first reading, it was not immediately clear to me what parts of the task were similar when assessing the Target and the Friend – only when reading the results did I understand that most questions regarding the Target were also posed for the Friend. If the authors decide to revise this section, it would only improve an already great paper.

In sum, I found this manuscript to be as close to be published in its current from as possible. For that, I congratulate the authors.”

We again thank the reviewer for their positive assessment of our paper. We have revised the manuscript to more clearly communicate that the same questions asked of the target were asked of the friend (see ll. 8).

“Next, immediately following ratings of the target, participants viewed a photo of a neutral male face (taken from the Radboud Faces Database, Langner et al., 2010) described as a close friend of the previous individual and were asked to estimate the same state and trait emotions (see Figure 1).”

---

## [Decision Letter · Decision Letter 1]

13 Dec 2024

PONE-D-24-25340R1Possessing Potential Weapons (Still) Heightens Anger Perception: Replicating and Extending a Test of Error Management TheoryPLOS ONE

Dear Dr. Moser,

I have received two independent reviews from experts in the field of emotion. I am very grateful for their rigorous evaluations. The reviewers have identified potential in your paper; however, they also raised several critical concerns and offered suggestions.

The comments from the reviewers are included below. Specifically, Reviewer 4 raised several critical concerns regarding the methodological rigor and transparency. These concerns shall be fully addressed through amendments, updates to results, additional analyses, or clarifications of limitations. Reviewer 5 also provided practical suggestions for enhancing clarity. I have independently reviewed your manuscript and concur that these concerns need addressing.

After careful consideration, I believe that your manuscript has merit, but in its current form, it does not fully meet the publication criteria of PLOS ONE. As the reviewers’ concerns provide clear direction for the revision of your manuscript, we invite you to submit a revised version that addresses the points raised during the review process.

We look forward to receiving your revised manuscript.

Kind regards,

June Chun Yeung

Academic Editor

PLOS ONE

Reviewers' comments:

Reviewer's Responses to Questions

**Comments to the Author**

1. If the authors have adequately addressed your comments raised in a previous round of review and you feel that this manuscript is now acceptable for publication, you may indicate that here to bypass the “Comments to the Author” section, enter your conflict of interest statement in the “Confidential to Editor” section, and submit your "Accept" recommendation.

Reviewer #4: (No Response)

Reviewer #5: All comments have been addressed

2. Is the manuscript technically sound, and do the data support the conclusions?

Reviewer #4: No

Reviewer #5: Yes

3. Has the statistical analysis been performed appropriately and rigorously? 

Reviewer #4: No

Reviewer #5: Yes

4. Have the authors made all data underlying the findings in their manuscript fully available?

Reviewer #4: No

Reviewer #5: Yes

5. Is the manuscript presented in an intelligible fashion and written in standard English?

Reviewer #4: Yes

Reviewer #5: Yes

6. Review Comments to the Author

Reviewer #4: The authors conducted a useful study consisting of a replication, extended to test a novel hypothesis. Their work constitutes valuable contributions to the field. However, while reading the manuscript I noticed some errors in the interpretation of the results, as well as doubts as to how the statistical analyses were conducted. I present them along with recommendations on how to improve the paper beyond its current state below. I realize that the authors have already passed through one stage of revisions and might feel tired with the revisions, however I believe that the points I’ve outlined below will improve the works scientific value and allow the readers to fully benefit from reading it. I encourage the authors to share my perspective.

1. , The authors test all of their hypotheses related to state and trait emotions both in the target and the friend condition using one huge MANOVA, controlling for the specific picture condition that was shown for the friend condition. This does not make sense as the ratings for the target were presented prior to presenting the friends faces, therefore the friend face condition cannot possibly modulate the responses of participants with regards to the target. The authors should rerun their analyses, separating the tests for the target, and the friend condition.

2. The authors should explicitly specify the type of replication they are running on the Holbrook et al., 2014. In my estimation this is a direct replication, as opposed to conceptual, however authors should double check it in light of their knowledge of the methodology of their study.

3. The authors write:

“However, departing from prior findings, we observed no effects of the object manipulation on trait fear, disgust, or dishonesty, potentially owing to the greater power in this study (N =476) compared to the initial study (N =264).”

This interpretation appears to misconstrue the concept of statistical power. Statistical power is defined as the probability of detecting an effect, assuming that an effect actually exists. Greater statistical power increases the likelihood of identifying a true effect, not the likelihood of failing to find one. Therefore, attributing the absence of effects to higher statistical power is conceptually inconsistent. A more plausible explanation might involve differences in study design, sample characteristics, or genuine null effects in this larger sample. This section should be revised.

4. The authors write:

“While this null finding may be interpreted as evidence against the proposed ”reverse-halo” effect with respect to appraisals of allies”

However, this interpretation is not justified. A null finding does not provide evidence against the hypothesis; it merely indicates that the test failed to detect an effect under the conditions of the study. Without a prior assumption about the expected effect size, the test is inconclusive and cannot substantiate or refute the validity of the hypothesis, nor inform its validity in any way. Aside from that, the two methodological limitations that the authors proceed to outline after the snippet above do not include the potential insufficient power limitation. This section should be revised.

5. I recommend the authors to specify the least meaningful effect size of interest for their reverse halo effect, conduct a post-hoc power analysis, and comment on it in the discussion. Additionally, since some of the Holbrook effects were not replicated despite having a twice as large sample, similar post-hoc power analyses would be in order for the replication effects.

6. It would be useful to expand on alternative or complementary explanations of the recovered effect for the knife condition. This meta-analysis https://journals.sagepub.com/doi/10.1177/1088868317725419 for example explains similar effects using the General Aggression Model (GAM), authors are free to take inspiration from it. (Disclaimer: I did not author it, nor am I personally benefiting from recommending it in any shape or form).

7. While I agree with the limitation of low ecological validity with regards to the model holding the knife, I would argue that the friend model photos you used are even less ecologically valid, given that they are basically disembodied faces, without traces of necks nor bodies, which might have directly prevented you from noticing the reverse halo effect. I recommend the authors to add this point to the limitation section, hoping that other researchers will test your hypothesis with more ecologically valid methods in the future, which the authors could also consider adding as a future studies direction recommendation. (I understand that the choice of the stimuli might have been dictated by the want to control information such as clothing etc., but this seems like an overkill)

8. I recommend the authors to specify that the study is a replication of the Holbrook study directly in the abstract so that meta-analysts will have an easier time finding it in the future. Failed (or partially replicated) published replications are just as useful to that end as they are rare.

9. Finally, it is my duty as a reviewer to ensure that the study has been conducted in accordance with ethical standards, especially since it is a preregistration. For that reason, I will kindly request the authors to explain why the csv file that they have used to store the results is named EMT2, implying that it could be a second version of a file, and why the initial R code uploaded on the same date as the preregistration was deleted, and replaced by a different R code. Furthermore, the R code that has been uploaded later uses a csv named EMT.csv, and not EMT2 implying that the csv was altered after the code has been run. Explanations for all of that would be most welcome.

10. Connected to the previous point, I wasn’t able to reproduce the same statistical results as reported in the paper using the code and data available on the osf platform. Granted, the results look similar – for example the computed F value for State Anger is 12.58, while authors report 13.51 in the paper. Reproducible results need to be provided and reported. Please also explain the reason for why this discrepancy currently exists.

11. When it comes to describing the statistical tests all that authors write is that they employed the general linear model. When I first read the paper, I understood from that that the authors conducted parallel regressions for each dependent variable, which led me to question whether presenting significant results without correcting for multiple comparisons is necessary. When I was finally able to see the code, I noticed that the authors used a MANOVA for their results, which is indeed the preferred analysis for this situation and eliminates the multiple comparison problem. This should be reported in more detail in the manuscript text.

12. Finally in the R code, the authors test the effect of the condition on state fear and state disgust, yet they do not report the results. This is concerning as the authors do not report any hypotheses with regard to those metrics, yet they include it in the manova. Why?

Overall, the two main issues I have with the manuscript currently is the lack of transparency with regard to how the analyses were conducted, as well as the inaccuracies in the interpretation of results. Unless these issues are corrected, I deem the manuscript unfit for publication. Also, until the reproducibility issues are solved I treat this situation as if the authors did not share all of the data they used to generate the results from the manuscript.

Reviewer #5: The study has two main objectives. Firstly, it aims to replicate Holbrook et al. (2014)’s work and examine whether a target in the armed condition (vs. unarmed condition) is perceived to be angrier. Secondly, it extends prior replication work by exploring whether a friend of the target in the armed condition (vs. unarmed condition) is also appraised as angrier as well.

With the sample size larger than that in Holbrook et al. (2014), the authors replicated the result that participants perceived the target holding a threatening (vs. non-threatening) object as angrier. However, the result was not extended to their friend of the target. Interestingly, the authors found that the affective homophily was present, as the state and trait emotion ratings of both the target and their friend were generally similar.

While I am not an expert in the field of emotion perception, I can understand most parts of the research report. Since this is my first time reading it, I believe the authors could make some slight revisions to enhance clarity, and I have the following suggestions:

#1) In line 78, it would be clearer if the authors replaced the word 'control' with 'unarmed,' assuming that the control condition refers to the unarmed condition. In line 79, the authors can also consider revising the text to clearly state whether the model in the control condition is the same as the model in the unarmed condition.

#2.1) In Figure 1 (Stimuli representing Target and Friend), the authors might consider adding labels for Models 1, 2, and 3 next to each photo of the target’s close friend. While reading the results of the post-hoc Tukey comparisons (lines 160-164), I was unsure which male photo represented Model 3. I assumed the male model with the shortest hair was Model 3.

#2.2) I also assumed the results of post-hoc Tukey comparisons (lines 160-164, and SI Table 1) refer to comparisons of state and trait emotions between friends in the armed condition.

#3) In line 166, the authors wrote ‘We ran multiple linear regressions on friend state and trait anger with both friend state and trait anger and condition as our predictor variables.’.

However, after checking with the data and the syntax, I believe the sentence should be revised to: ‘We ran multiple linear regressions on friend state and trait anger, using both the target’s state and trait anger and condition as our predictor variables.’.

7. PLOS authors have the option to publish the peer review history of their article (what does this mean?). If published, this will include your full peer review and any attached files.

Reviewer #4: No

Reviewer #5: No

---

## [Author Response · Author response to Decision Letter 2]

4 Feb 2025

Replies to the Editor

We thank the editor and reviewers for their helpful feedback on our manuscript. In particular, Reviewer 4’s care in checking the data and the pre-registration identified several methodological errors in the structure of our MANOVA design, which have now been addressed. While the effects of the current study remain qualitatively the same, several effect sizes shifted as a result of a difference between the code we registered to the OSF and the RMarkdown code we used to generate the manuscript. As a result of attempting to create cleaner tables, several effects were inadvertently left out, generating slightly different effect sizes. We explain these data issues further in our reply to Reviewer 4, below.

To correct these errors, we have updated our code in the OSF with the correct syntax. Additionally, we have checked and re-checked all tables in the manuscript to report the correct effect sizes for the study, and are now confident that human errors have been remedied.

----------------

Replies to Reviewer 4

R4.0: “The authors conducted a useful study consisting of a replication, extended to test a novel hypothesis. Their work constitutes valuable contributions to the field. However, while reading the manuscript I noticed some errors in the interpretation of the results, as well as doubts as to how the statistical analyses were conducted. I present them along with recommendations on how to improve the paper beyond its current state below. I realize that the authors have already passed through one stage of revisions and might feel tired with the revisions, however I believe that the points I’ve outlined below will improve the works scientific value and allow the readers to fully benefit from reading it. I encourage the authors to share my perspective.”

We do share Reviewer 4’s perspective and strongly appreciate the effort they invested in pointing out errors in our statistical analysis and interpretation. Thank you!

R4.1 “The authors test all of their hypotheses related to state and trait emotions both in the target and the friend condition using one huge MANOVA, controlling for the specific picture condition that was shown for the friend condition. This does not make sense as the ratings for the target were presented prior to presenting the friends faces, therefore the friend face condition cannot possibly modulate the responses of participants with regards to the target. The authors should rerun their analyses, separating the tests for the target, and the friend condition.”

Reviewer 4 is correct about this: in the OSF file, all of the effects were thrown into one large MANOVA, including models for the target (shown first) and the friend. The file we actually used to run the analysis of the data was an RMarkdown file which did, indeed, separate these by two models. The changes in the manuscript in this round of revisions therefore reflect alterations due to Reviewer comment 4.10, which we respond to below.

We have updated the text to reflect that we used two separate models with separate controls for the target model and the friend model:

Lls 147-152:

“To assess the effects of weapon manipulation on state and trait ratings of the target and the friend, we employed two separate general linear models on the target and friend measures in the form of MANOVAs to account for simultaneous multiple dependent variables. Separate from the target model, in the friend model we control for which friend's face was displayed.”

R4.2: “The authors should explicitly specify the type of replication they are running on the Holbrook et al., 2014. In my estimation this is a direct replication, as opposed to conceptual, however authors should double check it in light of their knowledge of the methodology of their study.”

R4 is correct that this is a direct replication with an extension. We clarify this in both the abstract, “Here, we conduct a direct replication of one such study examining these effects (Holbrook, 2014),” and in line. 52 of the manuscript, “we conducted a direct replication.”

R4. 3: “The authors write: ‘However, departing from prior findings, we observed no effects of the object manipulation on trait fear, disgust, or dishonesty, potentially owing to the greater power in this study (N =476) compared to the initial study (N =264).’

This interpretation appears to misconstrue the concept of statistical power. Statistical power is defined as the probability of detecting an effect, assuming that an effect actually exists. Greater statistical power increases the likelihood of identifying a true effect, not the likelihood of failing to find one. Therefore, attributing the absence of effects to higher statistical power is conceptually inconsistent. A more plausible explanation might involve differences in study design, sample characteristics, or genuine null effects in this larger sample. This section should be revised.”

The reviewer is correct that additional power, especially in the form of larger sample sizes, technically should not increase the likelihood of failing to find an effect. We have tightened up the language to state that the effects of the larger sample may have uncovered more precise effect estimates.

LLs 194-196

“The absence of effects in the larger sample may therefore indicate a more precise estimate of these effects and that the original findings, based on a smaller sample, could have been overestimated.”

R4.4: “The authors write: ‘While this null finding may be interpreted as evidence against the proposed “reverse-halo” effect with respect to appraisals of allies’

However, this interpretation is not justified. A null finding does not provide evidence against the hypothesis; it merely indicates that the test failed to detect an effect under the conditions of the study. Without a prior assumption about the expected effect size, the test is inconclusive and cannot substantiate or refute the validity of the hypothesis, nor inform its validity in any way. Aside from that, the two methodological limitations that the authors proceed to outline after the snippet above do not include the potential insufficient power limitation. This section should be revised.”

We agree with the reviewer here – our statement was not to note that we believe the null finding provides evidence against a hypothesis, but that other readers may interpret our results in this way. Additionally, we include language regarding the sample size limitation in both the methods (lls. 53-58) and the discussion:

LLs 212-215:

“First, while we doubled the size of the sample from the original Holbrook et al. (2014) study, a post-hoc power analysis indicated that a sample size of 784 observations would be necessary for detecting a small effect size, compared to the 476 obtained in this study.”

R4.5: “I recommend the authors to specify the least meaningful effect size of interest for their reverse halo effect, conduct a post-hoc power analysis, and comment on it in the discussion. Additionally, since some of the Holbrook effects were not replicated despite having a twice as large sample, similar post-hoc power analyses would be in order for the replication effects.”

We conducted post-hoc power analyses for the Holbrook et al. 2014 study, as well as of our present study with 6 outcomes, and of our paper with 9 outcomes in each model. For the Holbrook et al. study, a sufficient sample size of 683 was deemed sufficient for identifying a small effect size; for the present study, a sample size of 784 would have been sufficient. Our study therefore increased the power of the original Holbrook et al. study, bringing it closer to the necessary threshold for identifying small effect sizes, but possibly failed to identify true effects in our current study due to lower statistical power.

On the other hand, the strength and presence of the effects identified in both the Holbrook study and replicated in our study indicate that the true effect sizes may be small-medium, in which case the sample sizes needed would have been satisfied.

In addition to the text added in the discussion above (R 4.4), we now specify the post-hoc power analyses for both the original study and our replication.

LLs 53-58:

To evaluate the adequacy of our sample size, a post-hoc power analysis was conducted for the nine dependent variables representing ratings made for the friend across two conditions. The analysis was based on a small effect size (f0 = .02). The results of the post-hoc power analysis indicate that a total sample size of 784 observations is needed to achieve adequate statistical power for detecting a small effect.

R4.6: “It would be useful to expand on alternative or complementary explanations of the recovered effect for the knife condition. This meta-analysis https://journals.sagepub.com/doi/10.1177/1088868317725419 for example explains similar effects using the General Aggression Model (GAM), authors are free to take inspiration from it. (Disclaimer: I did not author it, nor am I personally benefiting from recommending it in any shape or form).”

We thank the reviewer for identifying the alternative literature on this condition. We agree that work on the General Aggression Model overlaps with aspects of our present research. However, the GAM is predominantly focused on inclinations for participants to aggress, rather than for appraisals of the emotional states of others. We discuss alternative interpretations, including GAM and priming effects, in the discussion now.

LLs 194-202:

“While similar hypotheses—such as semantic priming and the "general aggression model" (Benjamin et al., 2018)—suggest that associations with weapons should heighten general aggression tendencies, these effects were ruled out by controls in the original Holbrook et al. (2014) study. Fessler et al. (2023) replicated similar findings using both a knife and garden shears, further examining the relationship between lethal and non-lethal affordances in assessments of a target’s formidability.”

R4.7: “While I agree with the limitation of low ecological validity with regards to the model holding the knife, I would argue that the friend model photos you used are even less ecologically valid, given that they are basically disembodied faces, without traces of necks nor bodies, which might have directly prevented you from noticing the reverse halo effect. I recommend the authors to add this point to the limitation section, hoping that other researchers will test your hypothesis with more ecologically valid methods in the future, which the authors could also consider adding as a future studies direction recommendation. (I understand that the choice of the stimuli might have been dictated by the want to control information such as clothing etc., but this seems like an overkill).”

We thank the reviewer for this comment. The reason for introducing the photos as disembodied faces was in part to introduce the second-order formidability rating in the friend condition. Nevertheless, we agree that this may have been an influence on the results of the current study (as emotional appraisals are created with signals beyond those just appraised in facial expressions). We have acknowledged this in the revised discussion section:

LLs 229-233

“Relatedly, the friend stimuli, presented in the current study as one of three cropped neutral faces (done so to assess intuitions regarding their bodily formidability), were inherently less ecologically valid than the target stimuli, and may not have evoked emotion appraisals comparably to the presentation of the target's full body.”

R4.8: “I recommend the authors to specify that the study is a replication of the Holbrook study directly in the abstract so that meta-analysts will have an easier time finding it in the future. Failed (or partially replicated) published replications are just as useful to that end as they are rare.”

This is a good point, and we have now specified this in the abstract.

“Here, we conduct a direct replication of one such study examining these effects (Holbrook, 2014) and exploratorily test whether friends of an individual possessing a potentially lethal tool…”

R4.9: “Finally, it is my duty as a reviewer to ensure that the study has been conducted in accordance with ethical standards, especially since it is a preregistration. For that reason, I will kindly request the authors to explain why the csv file that they have used to store the results is named EMT2, implying that it could be a second version of a file, and why the initial R code uploaded on the same date as the preregistration was deleted, and replaced by a different R code. Furthermore, the R code that has been uploaded later uses a csv named EMT.csv, and not EMT2 implying that the csv was altered after the code has been run. Explanations for all of that would be most welcome.”

We very much appreciate the Reviewer’s commitment to both the pre-registration process and data integrity of our paper. The author is correct that EMT and EMT2 appear to be different files. Because the data were generated via Qualtrics, the original EMT file had column headings that were difficult to work with in R and were changed manually, creating the EMT2 file. These were meant to be changed locally to “EMT.csv”, but EMT2.csv was the file that was uploaded, hence EMT.csv appears in the OSF.

We have uploaded both the original Qualtrics File as “Qualtrics.csv” and the file with changed headings as EMT.csv for data clarity. The dates between the two should match and no alterations besides these were made.

R4.10: “Connected to the previous point, I wasn’t able to reproduce the same statistical results as reported in the paper using the code and data available on the osf platform. Granted, the results look similar – for example the computed F value for State Anger is 12.58, while authors report 13.51 in the paper. Reproducible results need to be provided and reported. Please also explain the reason for why this discrepancy currently exists.”

This was a syntax error in the coding that we were able to unravel after the reviewer helpfully identified it. The R file uploaded to the OSF used code that was part of an RMarkdown file which was used to generate the original manuscript. To format several of the tables, author CM removed several lines from the MANOVA (rather than silencing them in the table output). The data error is therefore reproducible using the code below:

cook.man.target <- manova(cbind(SA, TA, TF, TDisg,TDish,TU) ~

Condition*friend, data=d)

In this case, two things were happening. The target model was being conditioned on the friend, presented after the target, and there were fewer outcomes being measured and therefore slightly larger effect sizes. We have fixed this error and the data being reported now are those from the full model, which do not qualitatively alter any of the results, but indeed change several of the effect sizes. Again, we sincerely thank the reviewer for catching this error!

R4.11: “When it comes to describing the statistical tests all that authors write is that they employed the general linear model. When I first read the paper, I understood from that that the authors conducted parallel regressions for each dependent variable, which led me to question whether presenting significant results without correcting for multiple comparisons is necessary. When I was finally able to see the code, I noticed that the authors used a MANOVA for their results, which is indeed the preferred analysis for this situation and eliminates the multiple comparison problem. This should be reported in more detail in the manuscript text.”

We clarify this now in the text by specifying the model more precisely (see R4.1).

LLs 147-152:

“To assess the effects of weapon manipulation on state and trait ratings of the target and the friend, we employed two separate general linear models on the target and friend measures in the form of MANOVAs to account for simultaneous multiple dependent variables. Separate from the target model, in the friend model we control for which friend's face was displayed.”

R4.12: “Finally in the R code, the authors test the effect of the condition on state fear and state disgust, yet

---

## [Decision Letter · Decision Letter 2]

17 Mar 2025

PONE-D-24-25340R2Possessing Potential Weapons (Still) Heightens Anger Perception: Replicating and Extending a Test of Error Management TheoryPLOS ONE

Dear Dr. Moser,

Thank you for your thorough revision of the manuscript titled "Possessing Potential Weapons (Still) Heightens Anger Perception: Replicating and Extending a Test of Error Management Theory." I appreciate the effort you and your co-authors have put into addressing the reviewers' comments and clarifying methodological details.

The revisions successfully improve the methodological rigor and transparency of the study, particularly in response to concerns about statistical analyses and data reporting. Your responses to Reviewers 4 and 5 demonstrate careful attention to their feedback, and the modifications made to the manuscript effectively enhance its clarity and accuracy.

Upon my independent evaluation, I find that your manuscript is now close to a publishable standard. That said, there are still a few areas where minor, additional refinements are needed:

1. Currently, the manuscript does not provide a descriptive table summarizing the mean and standard deviation (SD) for all emotion ratings across all conditions (target and friend, armed vs. unarmed conditions). While the OSF script shows that some means and SDs were computed, not all emotion variables appear to be reported. I also noticed that some calculations for descriptive statistics (mean and SD) may be incomplete or inconsistent (https://osf.io/sdaqj, lines 70-90, Date modified: January 29, 2025). I recommend update the script and adding a descriptive statistics table to the results section, ensuring all reported outcome variables are included and clearly reported. For example, structuring a table to include rows for each emotion measure (e.g., state anger, trait anger, state fear, etc.), with separate columns for the experimental conditions (armed/unarmed) and for the target and friend ratings with Mean and SD.

2. Based on the descriptive table suggested in #1, I also recommend reporting the standardized differences and statistical tests (t-tests and Cohen’s d) between the armed vs. unarmed conditions. This would provide an initial insight into the experimental effects before the main MANOVA analysis.

3. Before conducting the main analysis, please examine and briefly report the key assumptions of MANOVA, including tests for multivariate normality and homogeneity of covariance matrices.

4. For all the MANOVA results (both main text and SOM), please report effect sizes (e.g., eta squared or omega squared) alongside F-values and p-values, and provide corresponding effect size interpretations.

5. In Table 3, the reported values are labeled as β (beta coefficients) and the caption is about regression analysis, but based on the provided description and analysis, these values appear to be correlation coefficients (r). If the analysis was conducted using correlation, the caption should be revised to reflect this. If the authors intend to report regression results (with more than one predictor), run the regression analysis accordingly and report the appropriate regression coefficients separately.

Given the above points, I am recommending Minor Revision for your manuscript. Please submit your revised manuscript by May 01 2025 11:59PM. If you will need more time than this to complete your revisions, please reply to this message or contact the journal office at plosone@plos.org. Please include the following items when submitting your revised manuscript:

We look forward to receiving your revised manuscript.

Kind regards,

June Chun Yeung

Academic Editor

PLOS ONE

Journal Requirements:

Reviewers' comments:

Reviewer's Responses to Questions

**Comments to the Author**

1. If the authors have adequately addressed your comments raised in a previous round of review and you feel that this manuscript is now acceptable for publication, you may indicate that here to bypass the “Comments to the Author” section, enter your conflict of interest statement in the “Confidential to Editor” section, and submit your "Accept" recommendation.

Reviewer #4: All comments have been addressed

2. Is the manuscript technically sound, and do the data support the conclusions?

Reviewer #4: Yes

3. Has the statistical analysis been performed appropriately and rigorously? 

Reviewer #4: Yes

4. Have the authors made all data underlying the findings in their manuscript fully available?

Reviewer #4: Yes

5. Is the manuscript presented in an intelligible fashion and written in standard English?

Reviewer #4: Yes

6. Review Comments to the Author

Reviewer #4: I applaud the authors on the hopefully final version of their article. All of the comments that I've had have been appropriately addressed.

7. PLOS authors have the option to publish the peer review history of their article (what does this mean?). If published, this will include your full peer review and any attached files.

Reviewer #4: **Yes:** Hubert Plisiecki

---

## [Author Response · Author response to Decision Letter 3]

27 May 2025

Comment to the Editor:

We thank the editor for their comments and feedback on our manuscript following our previous revision.

For data clarity and robustness purposes, the Editor requested us to report several additional measures from the data we obtained in our experiment, including the means and standard deviations of all measures in the study, independent t-tests for state and trait emotion comparisons, and effect sizes for all statistical tests. The Editor also requested that we examine the multivariate normality and homogeneity assumptions for our MANOVA.

We have updated the manuscript to now report these data, as well as the results from the MANOVA assumptions. These are now reported in a new Table 1 (which contains all means and standard deviations for measures in the study), a new Table S1 (which contains results from the independent t-tests between variables and conditions), and columns reporting effect size where appropriate (Tables 2 & 3 and S1). While we found that several assumptions for the MANOVA were not met, the robustness of the MANOVA holds given our relatively large sample size and balanced group sizes, and directly corroborates results from the independent t-tests.

We have also updated our code in the OSF with corresponding changes in this version of the manuscript.

Editor’s Comments

1. Currently, the manuscript does not provide a descriptive table summarizing the mean and standard deviation (SD) for all emotion ratings across all conditions (target and friend, armed vs. unarmed conditions). While the OSF script shows that some means and SDs were computed, not all emotion variables appear to be reported. I also noticed that some calculations for descriptive statistics (mean and SD) may be incomplete or inconsistent (https://osf.io/sdaqj, lines 70-90, Date modified: January 29, 2025). I recommend update the script and adding a descriptive statistics table to the results section, ensuring all reported outcome variables are included and clearly reported. For example, structuring a table to include rows for each emotion measure (e.g., state anger, trait anger, state fear, etc.), with separate columns for the experimental conditions (armed/unarmed) and for the target and friend ratings with Mean and SD.

We thank the Editor for their guidance and have added a new Table 1 which shows means and standard deviations for all state and trait emotions along with new text highlighting the table in the main body of the text.

Lls 159-162:

As an initial step, we examined descriptive statistics for all emotional and trait ratings by condition (see Table 1), followed by independent samples t-tests to assess differences between the armed and unarmed conditions (see SI Table 1 for full statistical results, including Cohen’s d).

2. Based on the descriptive table suggested in #1, I also recommend reporting the standardized differences and statistical tests (t-tests and Cohen’s d) between the armed vs. unarmed conditions. This would provide an initial insight into the experimental effects before the main MANOVA analysis.

We have conducted these analyses, finding convergence with the MANOVA results where raters were more likely to rate the state and trait anger and trait unpleasantness of the target model as higher in the armed condition than in the unarmed condition. We now report these outcomes in a new SI Table 1 and have reported findings from the independent t-tests in the text.

LLs 162-168:

We identified significant differences in state (Marmed = 4.36, 162 SD = 2.31; Munarmed = 3.64, SD = 2.00) and trait anger (Marmed = 5.36, SD = 1.79; 163 Munarmed = 5.02, SD = 1.61) of the target, as well as in trait unpleasantness (Marmed = 4.98, SD = 1.76; Munarmed = 4.31, SD = 1.49). These effects were statistically significant: state anger, t(457) = 3.64, p < .001, d = 0.34; trait anger, t(460) = 2.19, p = .029, d = 0.20; and trait unpleasantness, t(453) = 4.44, p < .001, d = 0.41. All effect sizes fell within the small to approaching moderate range.

3. Before conducting the main analysis, please examine and briefly report the key assumptions of MANOVA, including tests for multivariate normality and homogeneity of covariance matrices.

We have made the recommended additions, including the results of Mardia’s test and Box’s M. Critically, we found violations of Mardia’s test of multivariate normality for the both the target and the friend condition and violations of equality of covariance for the friend condition. Nevertheless, because our data has relatively large sample sizes and equal sized groups and the MANOVA results corroborate those with the independent t-tests, we believe the results MANOVA to be robust, which we discuss in the manuscript.

LLs 169-181:

Prior to conducting the MANOVA analyses, we evaluated whether the assumptions of multivariate normality and homogeneity of covariance matrices were met. Mardia’s test indicated significant violations of multivariate normality for both the target and friend rating variables (Target: skewness χ² = 380.50, p < .001; kurtosis z = 9.02, p < .001; Friend: skewness χ² = 725.75, p < .001; kurtosis z = 13.56, p < .001). However, Box’s M test for equality of covariance matrices revealed no significant difference in covariance structures across experimental groups for the target ratings (p = .141), suggesting that the assumption of homogeneity was met. In contrast, Box’s M test for the friend ratings was significant (p = .0003), indicating that the homogeneity of covariance matrices assumption may be violated in that set. While MANOVA is generally robust to moderate departures from multivariate normality and unequal covariances, particularly with balanced group sizes and sufficient sample size, these assumption checks should be considered when interpreting the results.

LLs 191-195:

Importantly, despite assumption violations, the MANOVA results mirrored the pattern observed in the t-tests, with the same three outcome variables—state anger, trait anger, and trait unpleasantness—emerging as statistically significant. This convergence of results across analytical approaches increases confidence in the robustness of the observed effects.

4. For all the MANOVA results (both main text and SOM), please report effect sizes (e.g., eta squared or omega squared) alongside F-values and p-values, and provide corresponding effect size interpretations.

The MANOVA effect sizes are now reported as partial eta squared in the revised text. We have also added descriptions of the effect sizes where appropriate.

LL 168 (with regards to independent t-tests):

All effect sizes fell within the small to approaching moderate range

LL189-191 (with regards to results of the MANOVA):

These effects correspond to small effect sizes, suggesting reliable but modest differences in perceived anger based on the presence of a weapon.

5. In Table 3, the reported values are labeled as β (beta coefficients) and the caption is about regression analysis, but based on the provided description and analysis, these values appear to be correlation coefficients (r). If the analysis was conducted using correlation, the caption should be revised to reflect this. If the authors intend to report regression results (with more than one predictor), run the regression analysis accordingly and report the appropriate regression coefficients separately.

The reporting of the beta coefficients was an error and should instead be Pearson’s correlation coefficients. We thought we updated this in the previous version of the manuscript, but those changes were not pushed. We have updated the table to reflect this change.

---

## [Editor Report · Decision Letter 3]

30 May 2025

Possessing Potential Weapons (Still) Heightens Anger Perception: Replicating and Extending a Test of Error Management Theory

PONE-D-24-25340R3

Dear Dr. Moser,

Thank you for your thorough response and for the detailed revisions made to your manuscript. We’re pleased to inform you that your manuscript has been judged scientifically suitable for publication and will be formally accepted for publication once it meets all outstanding technical requirements.

Kind regards,

June Chun Yeung

Academic Editor

PLOS ONE

---

## [Editor Report · Acceptance letter]

PONE-D-24-25340R3

PLOS ONE

Dear Dr. Moser,

I'm pleased to inform you that your manuscript has been deemed suitable for publication in PLOS ONE. Congratulations! Your manuscript is now being handed over to our production team.

Kind regards,

on behalf of

Miss June Chun Yeung

Academic Editor

PLOS ONE